# An efficient 3D column-only P300 speller paradigm utilizing few numbers of electrodes and flashings for practical BCI implementation

**Onur Erdem Korkmaz**[1☯]*, **Onder Aydemir**[2☯], **Emin Argun Oral**[3☯], **Ibrahim Yucel Ozbek**[3☯]

**1** Ispir Hamza Polat Vocational College, Atatürk University, Erzurum, Turkey, **2** Department of Electrical and Electronics Engineering, Karadeniz Technical University, Trabzon, Turkey, **3** Department of Electrical and Electronics Engineering, Atatürk University, Erzurum, Turkey

☯ These authors contributed equally to this work.
* onurerdem.korkmaz@atauni.edu.tr

**Data Availability Statement:** The dataset supporting the conclusions of this article is available in the Kaggle public repository (https://www.kaggle.com/onurerdemkorkmaz/3d-column-

## Abstract

The event related P300 potentials, positive waveforms in electroencephalography (EEG) signals, are often utilized in brain computer interfaces (BCI). Many studies have been carried out to improve the performance of P300 speller systems either by developing signal processing algorithms and classifiers with different architectures or by designing new paradigms. In this study, a new paradigm is proposed for this purpose. The proposed paradigm combines two remarkable properties of being a 3D animation and utilizing column-only flashings as opposed to classical paradigms which are based on row-column flashings in 2D manner. The new paradigm is utilized in a traditional two-layer artificial neural networks model with a single output neuron, and numerous experiments are conducted to evaluate and compare the performance of the proposed paradigm with that of the classical approach. The experimental results, including statistical significance tests, are presented for single and multiple EEG electrode usage combinations in 1, 3 and 15 flashing repetitions to detect P300 waves as well as to recognize target characters. Using the proposed paradigm, the best average classification accuracy rates on the test data are improved from 89.97% to 93.90% (an improvement of 4.36%) for 1 flashing, from 97.11% to 98.10% (an improvement of 1.01%) for 3 flashings and from 99.70% to 99.81% (an improvement of 0.11%) for 15 flashings when all electrodes, included in the study, are utilized. On the other hand, the accuracy rates are improved by 9.69% for 1 flashing, 4.72% for 3 flashings and 1.73% for 15 flashings when the proposed paradigm is utilized with a single EEG electrode (P8). It is observed that the proposed speller paradigm is especially useful in BCI systems designed for few EEG electrodes usage, and hence, it is more suitable for practical implementations. Moreover, all participants, given a subjective test, declared that the proposed paradigm is more user-friendly than classical ones.

only-p300-speller-paradigm-dataset) and in the Supporting Information files.

**Funding:** This work was supported by the Atatürk University Scientific Research Projects Coordination Unit with the project number: FOA-2018-6524. We received financial support to buy all equipment (computer, EEG device, armchair, LCD screen) used in this study. The funders had no role in study design, data collection and analysis, decision to publish, or preparation of the manuscript.

**Competing interests:** The authors have declared that no competing interests exist.

## Introduction

Brain computer interface (BCI) is a system that links the brain to a wide variety of electronic devices including computer, robotic arm, and mobile phone without using the peripheral neural system and muscles [1–4]. In general, it takes electroencephalography (EEG) signal as input and uses features including motor imagery [5–7], event-related P300 potentials [8–10] or steady-state visually evoked potentials [11–13]. The event-related P300 potentials are preferred in EEG-based BCI systems since they show common features for most of the subjects, emerge in a short time and are not contaminated by eye-movement artifacts. Therefore, high performance and subject independent BCI systems can be proposed in terms of classification accuracy (CA) and information transfer rate.

The very well-known P300-based BCI application was introduced by Farwell and Donchin, who developed a protocol whereby a subject could choose among 36 alphanumeric characters, presented in a 6x6 matrix in which a row or column was randomly intensified on a computer screen [14]. This procedure was called as row–column (RC) P300 speller paradigm. In one such widely used system, a user focuses his/her attention on the specific target character, which he/she would like to spell and mentally counts how many times the character flashes in the matrix. It should be noted that he/she ignores any flashes which do not match the target character. Many studies have been carried out to improve P300 speller systems either by proposing signal processing algorithms or by developing classifiers in different architectures. For example, Oralhan proposed a 3-dimensional input convolutional neural network (CNN) model to investigate P300 detection for higher CA [15]. In another study, Kar et al. developed a feature tuning algorithm for selection of Autoregressive Yule Parameter features of optimal lag-length corresponding to individual electrodes with an aim to increase the P300 based BCI system performance [16]. In another newly developed feature extraction technique-based study, Acevedo et al. evaluated different orthogonal decompositions based on the wavelet transform for feature extraction, as well as different filter, wrapper, and embedded alternatives for feature selection for single trial P300-based BCIs [17]. In a deep learning-based work, Ditthapron et al. proposed an event-related potential encoder network (ERPENet), and a multitask autoencoder-based model, that can be applied to any ERP-related tasks. They stated that the ERPENet combined CNNs and long short-term memory, in an autoencoder setup. Therefore, it simultaneously compresses the input EEG signal and extract related P300 features into a latent vector [18].

In addition to the above-mentioned methods, many studies have been conducted to improve P300 speller systems by proposing new paradigms, enhancing the P300 waveform differences between target and nontarget characters. In one such study, Lu et al. found that ERP amplitudes were significantly greater in the self-face than in the famous face spelling paradigm [19]. They recorded EEG signals from ten subjects and provided 5.3% average CA improvement as compared to the use of the famous face paradigm, which achieved 80% average classification accuracy. In another research, Qu et al. proposed a three-dimensional (3D) single character visual paradigm [20]. They acquired the EEG signals from twelve volunteers and showed that it provided an average enhancement of 4.9% in the performance of the classical two-dimensional (2D) single character P300 speller system, where the CA was calculated as 89.1%. Because their paradigm was single character, it can be said that intensifying all the characters took longer compared to row-column based flashing paradigm. In another newly proposed paradigm, Ramirez-Quintana et al. tested only column intensified based P300 speller system instead of the classical row-column based paradigm [21]. They took EEG signals from eight subjects and twenty characters were spelled for the training and fourteen for the test. Although they proposed column intensified based paradigm which was improved P300 speller

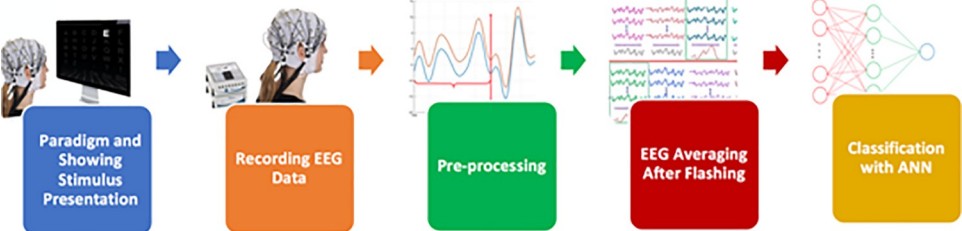

**Fig 1. Block-diagram of the proposed BCI system.**

system, they did not mention the increasing rate by comparing it with the classical RC P300 speller paradigm.

## Materials and methods

### The proposed method

In this study, we proposed a 3D column (3D-C) intensified-based P300 speller system, which provided a higher CA rate and lower user workload than the classical 2D row-column (2D-RC) P300 speller paradigm. The EEG signals were recorded in both 3D and 2D paradigm procedures from ten healthy participants. In both procedures, 60 characters were presented to each participant during the data collection phase. The proposed 3D-C paradigm was successfully applied to the datasets and we achieved an average CA rate of 99.81% for binary classification (target-nontarget) and 99.2% character detection accuracy on the test data by the traditional two-layer artificial neural networks (ANN) model with a single output neuron.

The proposed BCI system is composed of five main steps, and its block diagram is shown in Fig 1. Here, the proposed paradigm is presented to the subject on a LED display at the beginning. Corresponding brain electrical activities are collected simultaneously on an EEG cap, worn by the subject, to monitor the brain response of the subject to the stimulus, and they are recorded. After the EEG data collection and recording, EEG signals are averaged for each channel followed by a baseline removal step to obtain processed EEG signals with or without P300 waves depending on the flashings. Finally, target character classification is performed through use of an ANN utilizing the processed EEG data.

### Paradigm

BCI systems are generally constructed on the row-column paradigm model shown by Donchin et al. [22]. This traditional paradigm is based on the use of a 2-dimensional row-column-based (2D-RC) P300 spelling approach, which we refer as classical paradigm. In this study, a new 3D-C paradigm is proposed to better evaluate the visual P300 stimulus compared to the frequently used classical paradigm.

**Classical 2D paradigm.** In the classical 2D-RC paradigm, a 6x6 character matrix, as shown in Fig 2A, is presented to the participants. The participants are first instructed to focus

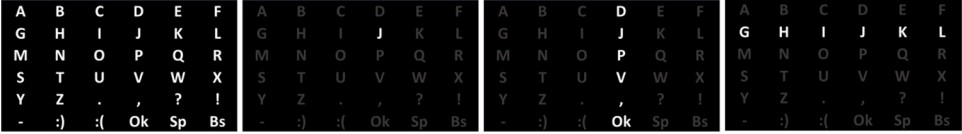

**Fig 2. Two-dimensional row-column (2D-RC) based classical paradigm screen shots on LED display.**

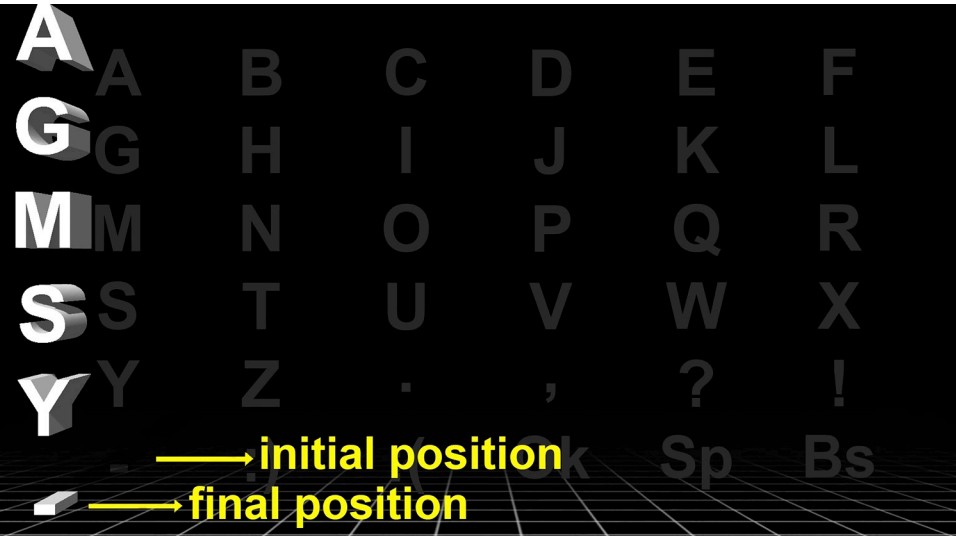

**Fig 3. 3D-C paradigm screen shots on LED display.**

on the specified character as shown in Fig 2B, and then columns and rows of the character matrix are flashed randomly (Fig 2C and 2D). They are also instructed to silently count the number of the target character flashings for a better concentration. A set of 12 flashings for each one of 6 rows of the character matrix, selected randomly, as well as for each one of 6 columns of the same matrix, also selected randomly, are referred to as "one trial". The classical paradigm employs 15 trials for each target character, referred to as "one run", and it consists of 180 (12x15) flashings. A total of 60 characters are shown to the participants as target and therefore, 60 runs are employed. The classical paradigm used in this work can be seen in the following link https://youtu.be/4GSpCtxuCIE

**The proposed 3D paradigm.** The proposed approach combines the methods of two paradigms given in the literature. These are the use of column-only based flashings suggested by Ramirez-Quintana et al. and 3D flashing of a single character suggested by Qu et al. Since the new paradigm involves both the use of column-only approach and 3D flashings, it is named as 3D-C paradigm. It has some major differences from the classical approach. First, after the columns of character matrix are flashed one after another, randomly selected rows are transposed and also flashed as columns to improve the stimulus. This idea is based on the fact that the amplitude of the P300 wave, using the columns, is greater than that of using rows because of the Western people's habit of reading texts horizontally [21,23]. On the other hand, it was shown by Orlandi et al. [24] that 3D visuals increase the ERP response of the brain. Based on this idea, a more efficient and faster BCI application can be designed to increase the ERP response. We also observed that paradigms utilizing 3D animations can further improve the stimulus to obtain improved ERP. Hence, the proposed paradigm combines the use of column-only flashing supported by 3D animations.

The new three-dimensional column-based visual paradigm sample LED display screen is shown in Fig 3, which shows the effect of 3D visualization. While the original 2D character set is displayed in the background with a faded color, a particular column with 3D effect is shown on the foreground with a bright color. It is clear from the figure that every character is re-positioned with a 3D perspective view. For example, the initial and final positions of character "-" are different and shown in this figure. The 3D animation effect, on the other hand, is accomplished in three steps. In the first step, the background character set is displayed for 75ms.

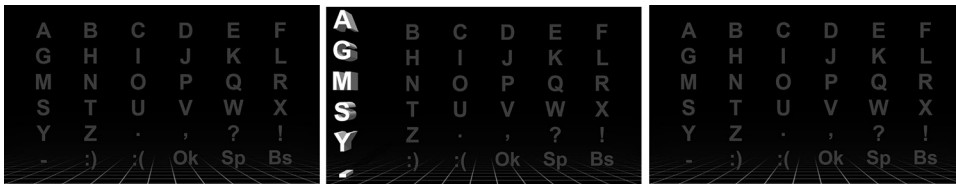

**Fig 4. Three-dimensional column-based experiment paradigm with duration.**

Then, the flashing column is displayed with 3D-shaped font characters for 100 ms on the foreground. Finally, the background character set is displayed again for 75ms. This 3-step animation is shown in Fig 4. The proposed paradigm used in this work can be seen in the following link https://youtu.be/mJbak5xPB7w

Another eye-catching feature of the 3D-C paradigm is the displaying of transposed rows as columns. In the beginning, all six columns are randomly flashed 15 times in the 3D manner as explained before. Hence, 90 column flashings take place before the row-transposing. Fig 5A1 shows the original character set, and Fig 5A2 and 5A3 show the column flashings screenshots for target character "J" as an example. For the following transposing step, the full target character is first transposed as shown in Fig 5B1, the target character is displayed for 5 seconds (Fig 5B2), and all columns are flashed randomly, giving a total of 90 flashings. A sample transposed column flashing with the target character is shown in Fig 5B3.

## EEG recording

The data collection process was approved by the Health Sciences Institute Ethics Board of Ataturk University, and all participants signed the Consent Form, verified by the board, before the start of the EEG recording session. In the study, all target stimuli were demonstrated using a 1920 x 1080 resolution LED display using the classical 2D-RC and 3D-C based visual P300 spelling paradigms. The experiments were conducted while the participants were sitting on a comfortable sofa positioned 1m from the screen. In both paradigms, the target character is displayed for 5s at the beginning of each run, composed of 180 flashings. Inter Stimulus Interval (ISI) between the flashings is set to 75ms as shown in Fig 6. Hence, each flashing is completed in 175ms. At the end of every 15-target character run in the 60-character experiment, a 3-minute break is given.

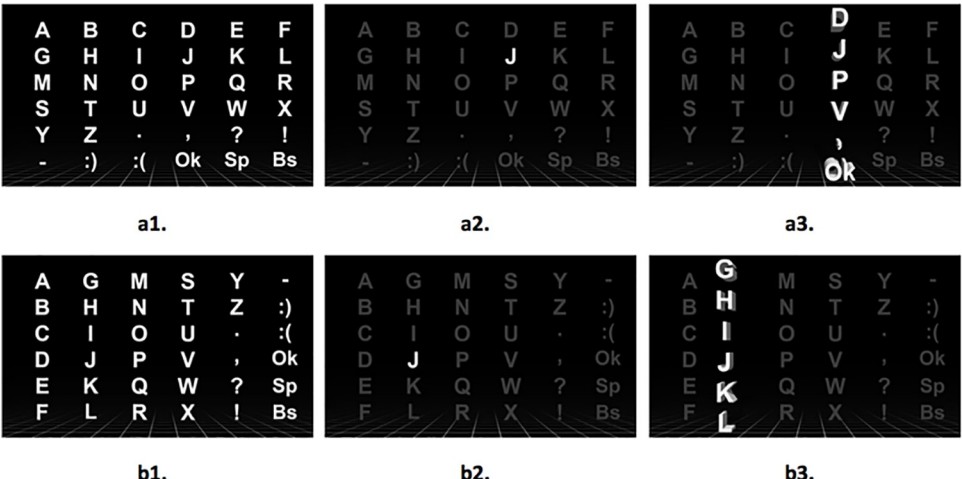

**Fig 5. Three-dimensional column-based experimental paradigm.**

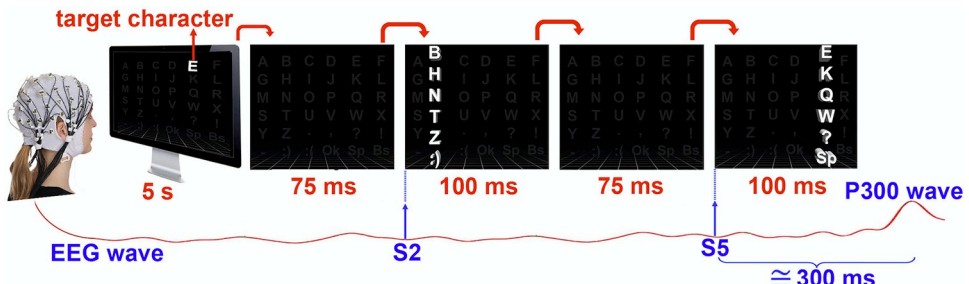

**Fig 6. 3D-C paradigm steps with corresponding durations.**

Two different experiments, one using 2D-RC and one using 3D-C, were carried out on two different days with a total of 10 participants, 5 men (mean age 28 ± 4.84) and 5 women (mean age 27 ± 4.15), to evaluate the performance of the proposed paradigm. In both experiments, EEG signals were captured using ActiChamp device by Brain Product. During the experiments, unipolar EEG recording was performed by placing the electrodes according to the international 10/20 system, and all electrode impedance values were maintained below 5 KΩ during data collection. The 'F2' channel was used as the reference electrode, and the sampling frequency for EEG recording was set to 250 Hz.

The data collection setup consists of an EEG device and different computers, one for stimulus presentation and one for data recording, as shown in Fig 7. A marker, which is the index of the row (or column) flashing and obtained from the stimulus computer, is combined with EEG recordings on the EEG device, and both are transferred to the recording computer. Thus, the EEG data and the visual stimulus index are synchronized as time-locked events.

## EEG signal processing

Signal processing is composed of two main steps as pre-processing and averaging. In the pre-processing step, EEG data, captured from 31 channels, is first passed through a band-pass filter

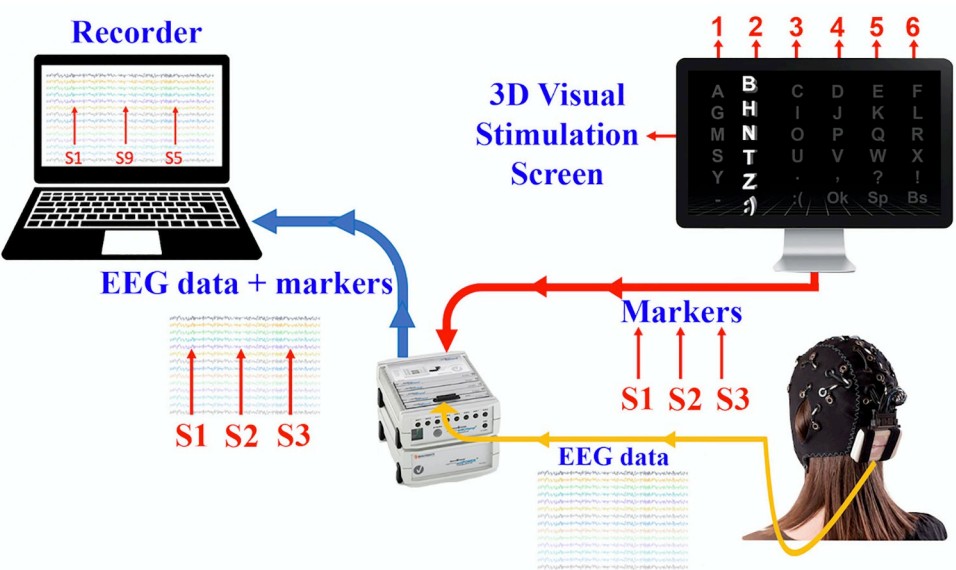

**Fig 7. EEG data and procedure for recording stimuli.**

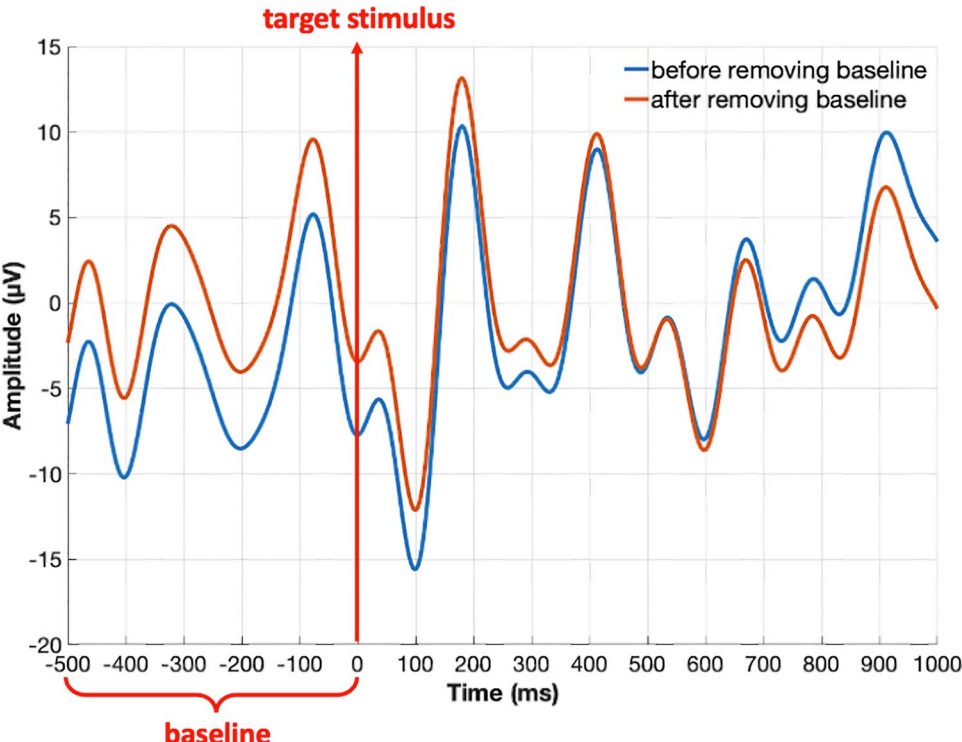

**Fig 8. Data after and before baseline noise removal.**

set to 0.1–10 Hz to remove noise as P300 waves have low frequency components [25]. Then the data is split into partitions with a single row (or column) marker. That is, each partition represents the EEG data for a specific flashing. Finally, a baseline-noise removal is applied to each partition. For that, the last 200ms long EEG data before the arrival of the target stimulus is used to determine the baseline, and it is used to correct the following 1000ms data after the stimulus. A sample data before and after the baseline removal along with -200ms to +1000ms time-interval of onset of the flashing is shown in Fig 8.

In the averaging step, on the other hand, partitioned EEG data are averaged as the P300 wave cannot be obtained clearly in a single trial because of its low SNR. For this purpose, all 15 flashings for each one of 12 rows and columns are averaged. That is, 12 averaged flashings are obtained as shown in Fig 9. This example was obtained from EEG data collected for the target character "J", located in the 4th column and 2nd row of the character matrix. It is clear that only the averages obtained for these particular row and column reveal the P300 waves while all others represent noise-like terms.

## Classification procedure

In all studies, target characters are presented to the participants during the data collection phase. Then, P300 wave detection in the pre-processed recorded samples is treated as a binary classification problem with "samples with P300" and "samples without P300" classes, and this categorization was performed by ANN.

For that purpose, a traditional two-layer ANN model with a single output neuron is used. The number of hidden layers and the number of neurons define the complexity of the neural network. In this study, a single hidden layer with M = 50 neurons defines the ANN model, and it is shown in Fig 10.

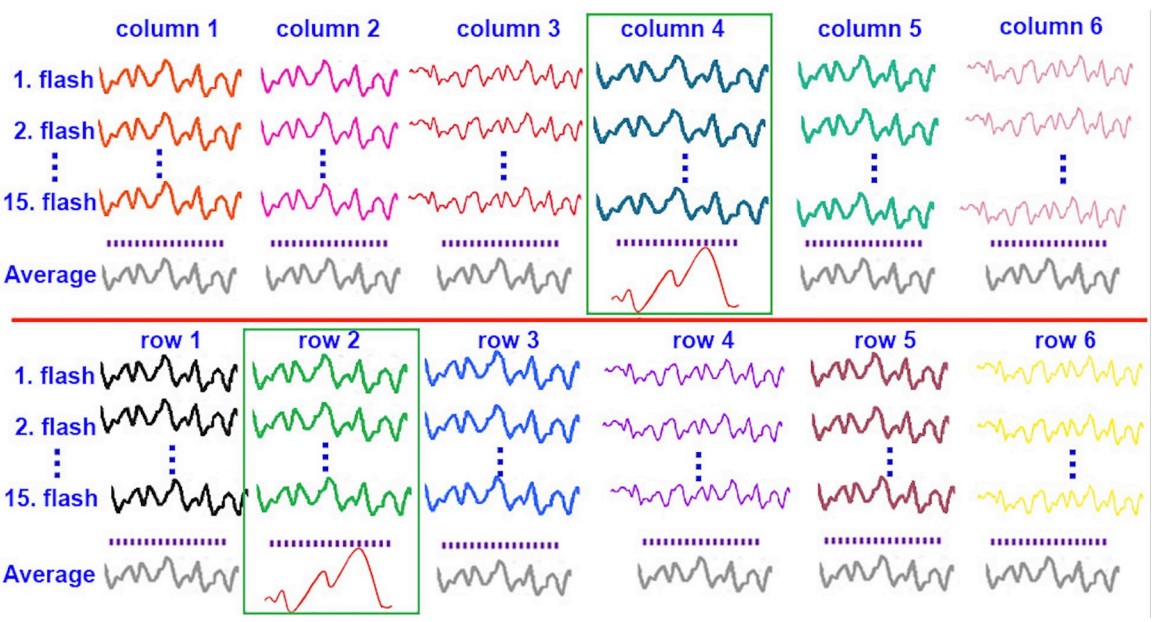

**Fig 9. P300 wave obtained by averaging the flashing.**

The corresponding two-layer neural network model with single output neuron is expressed as

$$\hat{y} = \tilde{g}\left( \sum_{j=1}^{M} {w_{1j}}^{(2)} * g\left( \sum_{i=1}^{d} {w_{ji}}^{(1)} * x_i + {w_{j0}}^{(1)} \right) + {w_{11}}^{(2)} \right) \tag{1}$$

where $x_i$ defines the $i^{th}$ input sample, $w_{ji}^{(k)}$ is the layer weight that connects the $i^{th}$ neuron at the

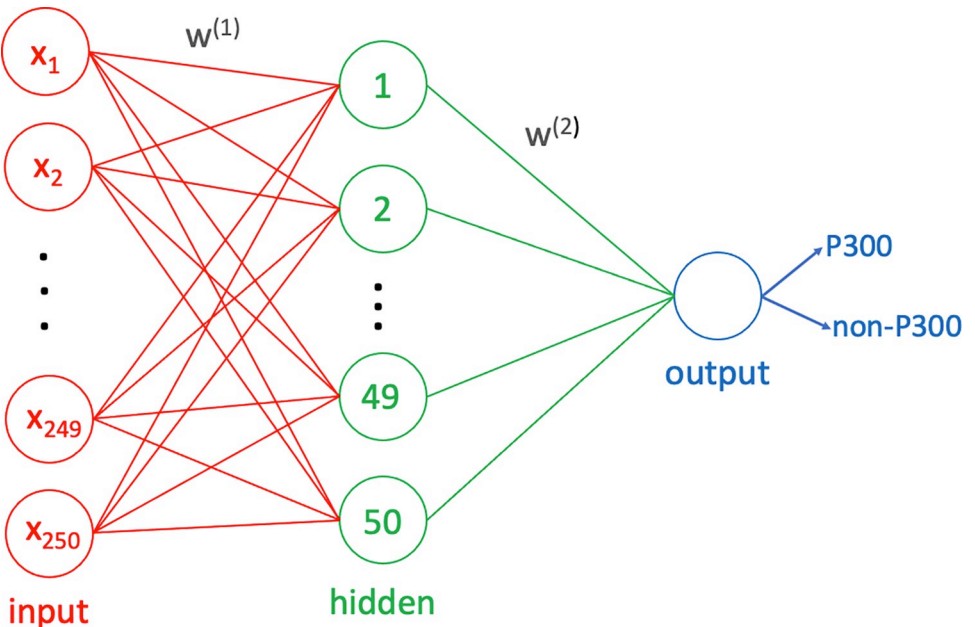

**Fig 10. Two-layer artificial neural network model.**

$k^{th}$ layer to the $j^{th}$ neuron in the next layer, $g$ is the tan-sigmoid function and $\tilde{g}$ is the linear function. Also, $d$ represents the dimension of the input vector, and it is 250 in the current study for 1-second-long EEG signals captured with $f_s$ = 250 sampling frequency. The corresponding total error over the whole dataset is defined as

$$J(w) = -\frac{1}{N}\sum_{n=1}^{N}[y_n log\hat{y}_n + (1 - y_n)\log(1 - \hat{y}_n)] \qquad (2)$$

where $N$ is the number of samples in the data set, $\hat{y}_l$ is the predicted value calculated by the neural network model for the $l^{th}$ dataset sample and $y_l$ is the sample label.

In the ANN model, the decision is performed in terms of the weights that are updated through the training step using the dataset samples. This update is performed by finding the new weights that minimize the error function, the difference between the estimated values and the desired values, as given in Eq 2.

## Experimental framework

In the proposed study, the binary classification problem (P300 or non-P300) was implemented using the EEG signals obtained from 10 participants in two different sessions to show the effectiveness of the proposed paradigm. In the first session, for this purpose, the classical 2D row column-based visual stimulus paradigm (2D-RC) was used while the proposed 3D-C was used in the second session, all with the same participants. In both sessions, the EEG data were collected on 31 channels as shown in Fig 11. This figure represents international standardized 10/20 system for EEG electrode positions.

These 31 EEG channels are grouped based on their location over the subject head as shown in this figure. Of these groups Central, Parietal and Occipital region 11 electrodes, marked with different colors in Fig 11, are used in this study. Central electrodes (C) are referred to as Cz, CP1, CP2; Parietal electrodes (P) are referred as P7, P3, Pz, P4, P8, and Occipital electrodes (O) are referred to as O1, Oz, O2. In the numbering of these electrodes, the odd numbers are used for left hemi-sphere, the even numbers are used for right hemi-sphere while sub-index "z" denotes the midline.

**Presentation of experimental results.** All experimental results of the proposed study are presented for two cases, namely P300 detection and target character recognition. In the

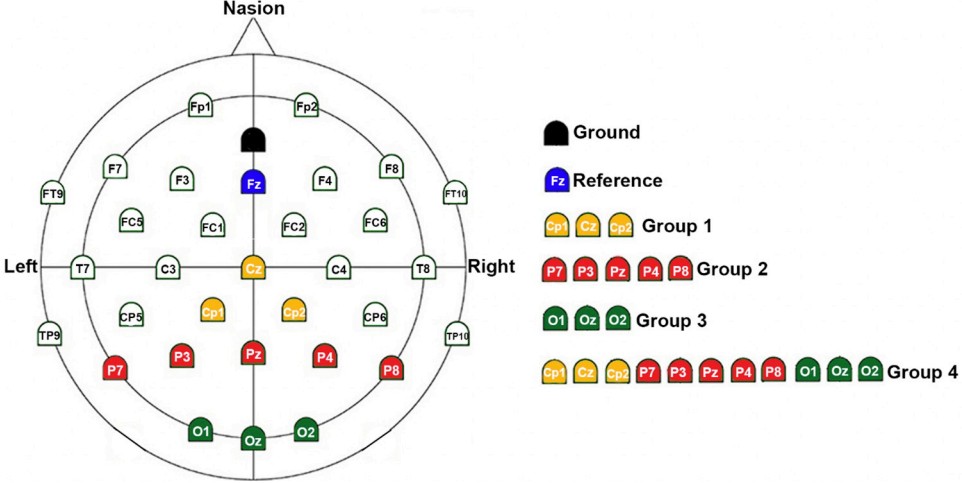

**Fig 11. Channels analyzed and their combinations.**

former, the existence of the P300 wave is examined following a character flashing, while in the latter, two sequential P300 waves are examined for column flashing and row flashing (or transpose of row for the proposed paradigm) to recognize the target character. The corresponding experimental results, presented in the next section, are organized based on two main parameters. These are the EEG electrode combination and the number of flashings for each participant subject.

In terms of the EEG electrode combination, various types of results (11 individuals and 4 groups, shown in Fig 11) are presented to define the effectiveness of the EEG electrode use as follows.

- 11 individual electrode results (Cz, CP1, CP2, P7, P3, Pz, P4, P8, O1, Oz, O2 separately)

- The result for Group 1 of 3 Central electrodes (combined Cz, CP1, and CP2)

- The result for Group 2 of 5 Parietal electrodes (combined P7, P3, Pz, P4, and P8)

- The result for Group 3 of 3 Occipital electrodes (combined O1, Oz, and O2)

- The result for Group 4 (combined Cz, CP1, CP2, P7, P3, Pz, P4, P8, O1, Oz, and O2)

In terms of the used number of flashings, on the other hand, 3 different results are presented as follows.

- Result for a single-flashing

- Result for 3 flashings

- Result for 15 flashings

In this study, results are presented using a radar plot as shown with a sample, given in Fig 12. Each radar plot consists of five nested pentadecagons of different circumcircles with radiuses 60, 70, 80, 90, 100, representing the average accuracy of the corresponding results. Parameter, for which the average accuracy is given, is located on the corners of the pentadecagon. In this example, the radar plot shows that 96.03% average accuracy is obtained for Oz electrode, and similarly, 99.81% average accuracy is obtained for Group 2 (combined P7, P3, Pz, P4, and P8) electrodes.

Also, considering the fact that the experiments are conducted with 10 different participants, all experimental results are presented using radar plots in terms of "EEG electrode combination", "used number of flashings" and "participant" parameters to better visually demonstrate the results.

**Performance measures.** In this section, the performance metrics, used in this study, are briefly defined. These are Accuracy, Area Covered by Radar Curve (ACRC), and Significance Test performance metrics.

*Classification accuracy.* Two types of accuracy metrics are obtained in this study. The first one defines the percentage of the correctly detected P300 wave, and the second one defines the percentage of the correctly recognized target character. Both accuracy terms are defined as the percentage of correctly classified instances as follows:

$$Accuracy = (TP + TN)/(TP + TN + FP + FN) \tag{3}$$

where **TP**,**FN**,**FP**, and **TN** represent the number of true positives, false negatives, false positives, and true negatives, respectively.

*Area Covered by Radar Curve (ACRC).* The area under a pentadecagon curve which was inspired by Aydemir [26], is also used as another performance metric to combine multiple accuracy rates distributed over the radar curves such as Fig 12. The ACRC is composed of

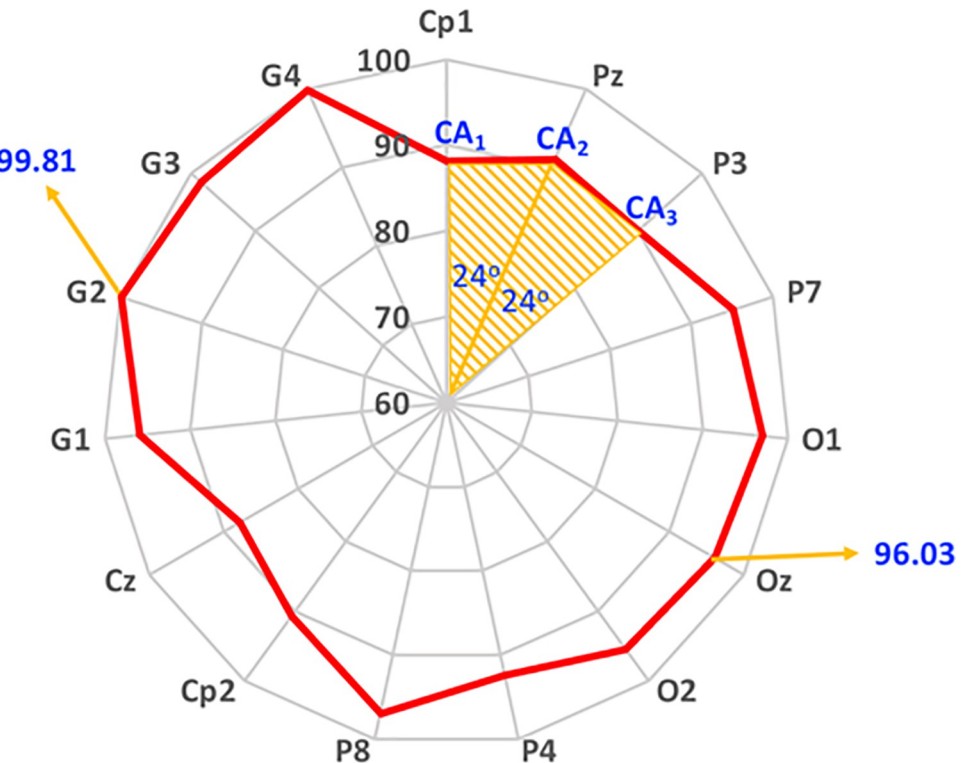

**Fig 12. Area covered by radar curve.**

triangular areas, defined by two accuracy values and a central angle—all located at the pentadecagon origin. Hence, ACRC is obtained as the total area of each triangular area as given in Eq 4. It should be noted that while the ACRC value is normalized between 0 and 1, the maximum value 1 and the minimum value 0 indicate the best and the worst situations, respectively.

$$AURC = \sum_{i=1}^{N} 0.5 * CA_i * CA_{i+1} * \sin(\alpha) \tag{4}$$

where α is for 24° for 15 parameters (electrode usage), all distributed over the pentadecagon circumference. Furthermore $CA_i$ represents the $i^{th}$ classification value, and N = 15 is the number of parameters in radar plot.

*Significance test*. The significance of the difference between two classification performances is also used as another metric to compare the CA results of the classical and the proposed paradigms. For that, a two-sample t-test of the null hypothesis of the similarity is compared with a degree of freedom of 9 and a threshold value of 0.05 and 0.01.

**Training and test data preparation.** The dataset records are divided into 1-second length EEG data using a sampling frequency of 250 Hz, captured on each one of 31 EEG channels. This dataset is split into training, validation, and test sets with 70, 15, and 15 percent, respectively, to be used in the binary classification step. In this study dataset splitting, and tests are performed twenty times, and all relevant CA results are obtained by averaging the corresponding twenty runs to obtain more reliable test results.

All training and test algorithms are utilized in the MATLAB R2018a environment on a 2.4 GHz Intel Core i5 processor-powered computer with 8GB, 1600 MHz DDR3 memory. To present the performance of the proposed method, two different test results are performed, i.e., P300 wave detection and target character recognition.

## Results

In this study, we proposed an efficient 3D column-only P300 speller paradigm utilizing a few numbers of electrodes and flashings for practical BCI implementation. In this section, we provided the experimental results including P300 detection and the target character recognition performances achieved by the traditional two-layer ANN model with a single output neuron.

### P300 detection results

The obtained CA results in terms of subjects, specific electrodes as well as electrode groups are presented as radar plots in Figs 13 and 14. All results in both figures are presented for 1, 3, and 15 flashings both for the classical paradigm in color blue and for the proposed paradigm in color red to reveal the performance change with the number of flashing repetitions.

Considering Figs 13 and 14, it appears that the 3D-C paradigm provides roughly higher CA for 1, 3, and 15 flashings than the 2D-RC paradigm. In other words, blue lines are encapsulated by red lines in the radar plots except for specific subjects, different flashing repetitions, or channels.

For a precise assessment, on the other hand, considering various single/multiple EEG electrode usage combinations, the overall performance can be measured in terms of ACRC as it involves the CA rates for all single electrodes as well as their different combinations. Hence, the ACRC values of each radar plot of Figs 13 and 14 are given in Table 1 for a detailed evaluation. Based on the mean ACRC values for all subjects, given at the bottom of Table 1, the performance increases with the use of the proposed paradigm for all flashing values.

Moreover, the overall performance of the BCI system improves to the ACRC values of 0.67, 0.79, and 0.91 for 1, 3, and 15 flashings, respectively, with the use of the proposed paradigm. Namely, the average ACRC values of all subjects increase by 15.5%, 6.75%, and 2.24% for 1, 3, and 15 flashings, respectively, with the use of the proposed paradigm.

Considering the single EEG electrode usage, it is observed that the P8 electrode gives the highest accuracy rate as can be seen from Figs 13 and 14. The corresponding accuracy rates for single P8 electrode use are given in Table 2. In this table, the CA performance measure is considered instead of the ACRC since a particular electrode usage is considered. Here, the overall performance of the BCI system improves to the CA values of 86.37%, 92.23%, and 97.83% for 1, 3, and 15 flashings, respectively, with the use of the proposed paradigm. That is to say, the classical performance values increase by 9.69%, 4.72%, and 1.73% for 1, 3, and 15 flashings, respectively.

Considering the multiple EEG electrode usage, it is observed that Group 4 electrodes give the highest accuracy rate as can be seen from Figs 13 and 14. The corresponding accuracy rates for Group 4 electrodes are given in Table 3. Here, the overall performance of the BCI system improves to the CA values of 93.90%, 98.10%, and 99.81% for 1, 3, and 15 flashings, respectively, with the use of the proposed paradigm. That is, the classical performance values increase by 4.36%, 1.01%, and 0.11% for 1, 3, and 15 flashings, respectively.

### Target character recognition results

The character recognition tests are performed using 3-fold cross-validation by which 60 target characters are partitioned into three sets and one set is used for training while the remaining two are used for testing. All presented test results were obtained for all three different partitions. Also, all character identification tests are performed in both paradigms for 1, 3, and 15 flashings using only Group 4 electrodes as it gives the best P300 detection performance. The corresponding results are shown in Table 4. Each entry in this table contains correctly detected character numbers out of 40 target characters as well as its percent ratio. As an example, the entry for the 2D paradigm with 1 flashing for Subject1 represents 23 correct character detections, corresponding to 57.5% of target character detection accuracy.

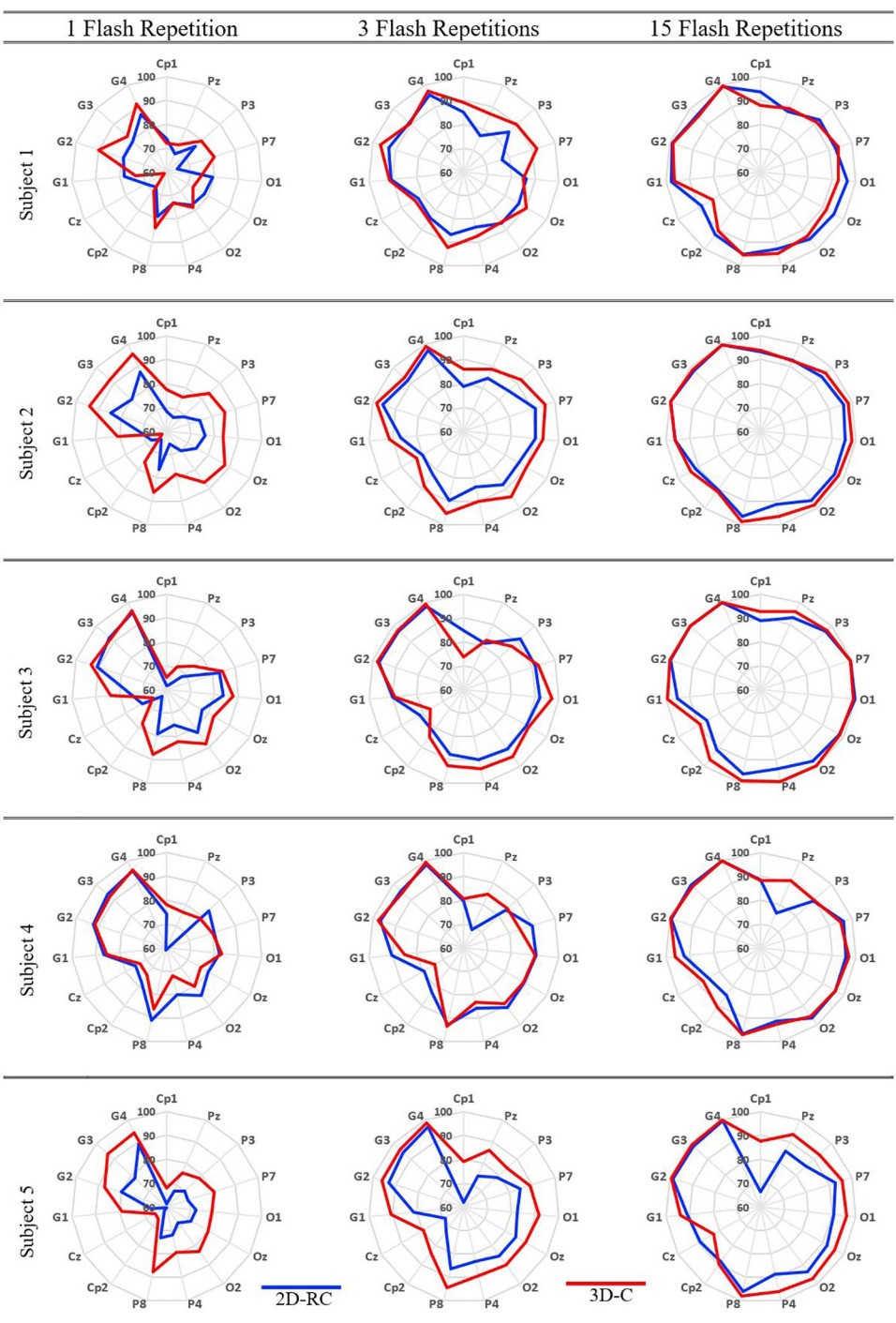

**Fig 13. Different repetitions result for subjects 1-2-3-4-5.**

## Subjective test results

In this study, a subjective test was given to all participants in order to evaluate the workload of the proposed paradigm. The questions and the answers of all subjects are given in Table 5. Based on these answers, it can be concluded that the proposed paradigm not only provides higher CA performance but also relieves the workload on the participants.

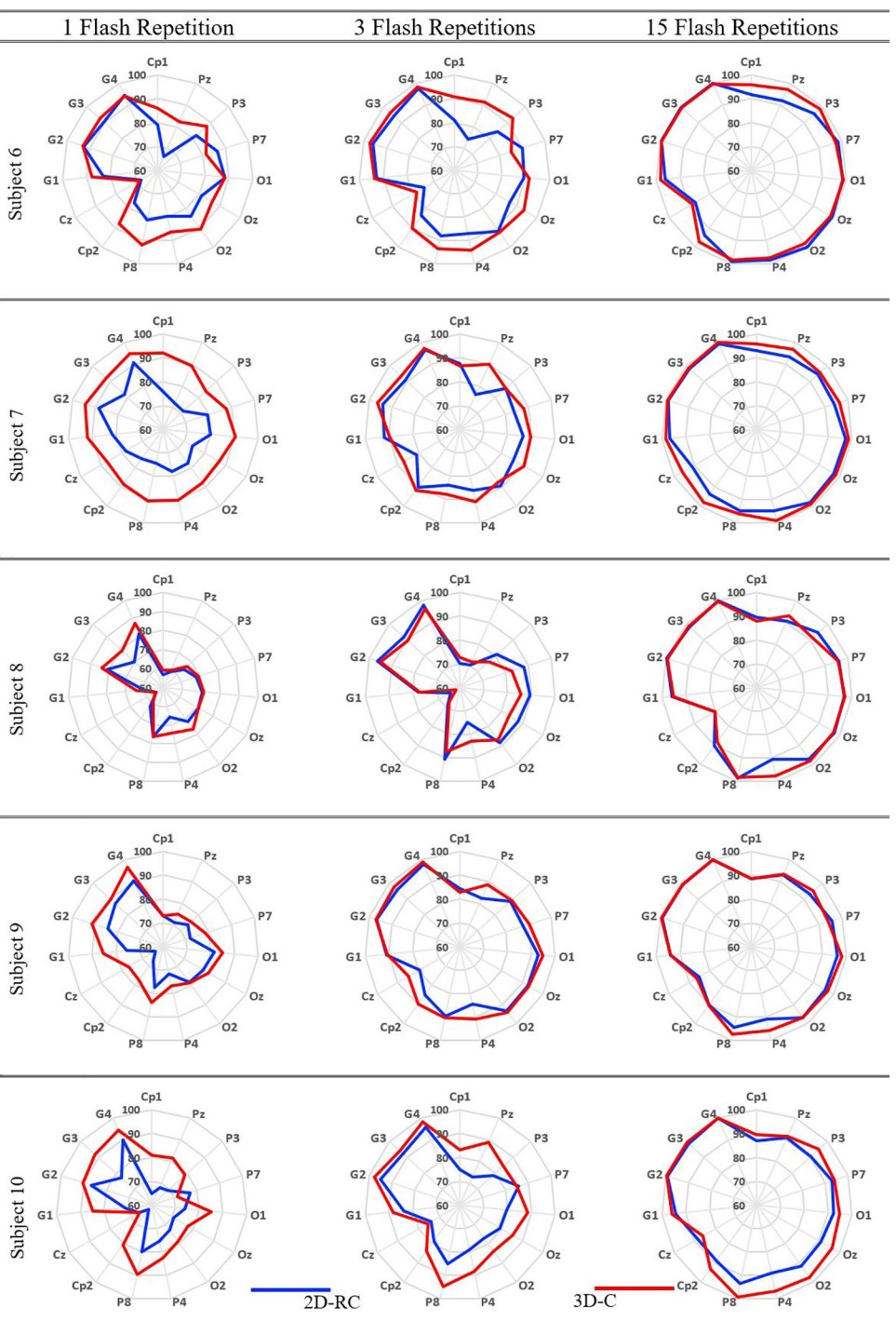

**Fig 14. Different repetitions result for subjects 6-7-8-9-10.**

## Discussion

In the previous section, the performance of the proposed paradigm is compared with the classical paradigm in terms of various aspects. First, the performance of every single subject is evaluated by means of ACRC for all electrode usage combinations. Second, the single electrode use

**Table 1. ACRC values of radar plots for Figs 13 and 14.**

| | 1 flashing | | 3 flashing | | 15 flashing | |
|---|---|---|---|---|---|---|
| | 2D-RC | 3D-C | 2D-RC | 3D-C | 2D-RC | 3D-C |
| Subject1 | 0.57 | 0.59 | 0.74 | 0.80 | 0.89 | 0.86 |
| Subject2 | 0.53 | 0.69 | 0.76 | 0.84 | 0.90 | 0.93 |
| Subject3 | 0.60 | 0.68 | 0.81 | 0.81 | 0.91 | 0.95 |
| Subject4 | 0.68 | 0.67 | 0.77 | 0.76 | 0.86 | 0.89 |
| Subject5 | 0.50 | 0.64 | 0.67 | 0.80 | 0.82 | 0.89 |
| Subject6 | 0.68 | 0.76 | 0.76 | 0.84 | 0.93 | 0.95 |
| Subject7 | 0.62 | 0.81 | 0.76 | 0.81 | 0.91 | 0.95 |
| Subject8 | 0.46 | 0.49 | 0.68 | 0.66 | 0.89 | 0.89 |
| Subject9 | 0.59 | 0.67 | 0.80 | 0.84 | 0.89 | 0.90 |
| Subject10 | 0.54 | 0.68 | 0.68 | 0.77 | 0.86 | 0.91 |
| **Mean** | **0.58±0.072** | **0.67±0.086** | **0.74±0.051** | **0.79±0.054** | **0.89±0.031** | **0.91±0.031** |

performance is assessed by means of CA. Finally, as the third case, multiple electrode use is also considered for maximum accuracy performance. It is observed that the performance of the proposed paradigm improves that of the classical paradigm for all these three usages. Moreover, it is informative to investigate the statistical significance of the obtained results. For that purpose, a significance test is performed for the three cases, namely performance of ACRC value, the performance of single electrode usage, and performance of multiple electrode usage. The corresponding results are given in Fig 15 in terms of the number of flashings. It is clear that improvements in ACRC value performance and single electrode usage results are statistically significant. On the other hand, the accuracy improvement with the use of the proposed paradigm for multiple electrode usage is significant for 1 and 3 flashings while it is below the alpha level since the use of all electrodes with 15 flashings in the classical paradigm already gives a high accuracy rate of 99.7%. Although the proposed paradigm slightly improves this accuracy rate up to 99.8%, it is not significant. Based on the obtained significance test results, it is suggested to employ the proposed paradigm, especially with less number of electrodes and flashings. Considering the fact that an easily implemented BCI system requires less number of electrode and flashing usage as well as a high level of accuracy, the proposed paradigm is preferable compared to the classical paradigm.

**Table 2. P8 electrode accuracy rates.**

| P8 | 1 Flashing | | 3 Flashing | | 15 Flashing | |
|---|---|---|---|---|---|---|
| | 2D-RC | 3D-C | 2D-RC | 3D-C | 2D-RC | 3D-C |
| Subject1 | 79.03 | 83.86 | 86.72 | 92.31 | 95.03 | 95.61 |
| Subject2 | 76.39 | 86.17 | 89.72 | 95.03 | 96.50 | 98.78 |
| Subject3 | 78.97 | 87.81 | 87.69 | 92.75 | 96.22 | 99.11 |
| Subject4 | 90.83 | 86.03 | 92.86 | 93.47 | 96.94 | 96.97 |
| Subject5 | 73.06 | 87.61 | 86.44 | 94.61 | 96.06 | 98.03 |
| Subject6 | 81.14 | 92.06 | 87.92 | 93.48 | 99.17 | 98.58 |
| Subject7 | 74.44 | 90.61 | 83.83 | 87.83 | 94.78 | 96.25 |
| Subject8 | 76.11 | 76.03 | 90.78 | 87.53 | 98.75 | 98.33 |
| Subject9 | 77.39 | 84.00 | 89.72 | 90.44 | 94.50 | 97.36 |
| Subject10 | 80.06 | 89.56 | 85.03 | 94.81 | 93.72 | 99.31 |
| **Mean** | **78.74 ±4.93** | **86.37 ±4.51** | **88.07 ±2.74** | **92.23 ±2.75** | **96.17 ±1.78** | **97.83 ±1.24** |

**Table 3. Group 4 electrodes accuracy rates.**

| GROUP4 | 1 Flashing | | 3 Flashing | | 15 Flashing | |
|---|---|---|---|---|---|---|
| | 2D-RC | 3D-C | 2D-RC | 3D-C | 2D-RC | 3D-C |
| Subject1 | 86.44 | 91.36 | 95.44 | 97.25 | 99.25 | 99.58 |
| Subject2 | 87.25 | 95.58 | 97.06 | 98.86 | 99.72 | 99.64 |
| Subject3 | 95.39 | 95.97 | 98.31 | 99.17 | 99.83 | 99.94 |
| Subject4 | 95.56 | 95.64 | 98.28 | 99.11 | 99.92 | 99.89 |
| Subject5 | 88.81 | 93.92 | 96.75 | 98.47 | 99.61 | 99.81 |
| Subject6 | 94.42 | 93.94 | 97.42 | 98.15 | 99.78 | 99.73 |
| Subject7 | 90.64 | 94.72 | 96.50 | 97.03 | 99.33 | 99.81 |
| Subject8 | 81.08 | 86.78 | 97.89 | 96.06 | 100.00 | 99.75 |
| Subject9 | 90.33 | 96.64 | 97.75 | 98.81 | 99.81 | 99.86 |
| Subject10 | 89.75 | 94.50 | 95.72 | 98.14 | 99.81 | 99.89 |
| **Mean** | **89.97 ±4.49** | **93.90 ±2.90** | **97.11 ±1.01** | **98.10 ±1.02** | **99.70 ±0.24** | **99.81 ±0.12** |

In character detection, on the other hand, the correct detection of P300 both for column and for row (or transposed column) is necessary. Therefore, the accuracies of Table 4 are slightly lower than those of Table 3. Similarly, to the P300 wave detection, the proposed paradigm also gives better results than the classical paradigm for character detection. To further investigate the performance of the proposed paradigm in character detection, the significance test is performed for the character test experiments, and the corresponding results are given in Fig 16. Again, the proposed paradigm gives significantly better results in less number of flashings.

**Table 4. Character detection accuracies obtained for both paradigms.**

| Subject | 1 Flashing | | 3 Flashing | | 15 Flashing | |
|---|---|---|---|---|---|---|
| | 2D-RC | 3D-C | 2D-RC | 3D-C | 2D-RC | 3D-C |
| Subject1 | 23/40 (57.5%) | 25/40 (62.5%) | 29/40 (72.5%) | 33/40 (82.5%) | 40/40 (100%) | 40/40 (100%) |
| Subject2 | 20/40 (50%) | 25/40 (62.5%) | 31/40 (77.5%) | 36/40 (90%) | 40/40 (100%) | 39/40 (97.5%) |
| Subject3 | 26/40 (65%) | 27/40 (67.5%) | 38/40 (95%) | 36/40 (90%) | 40/40 (100%) | 40/40 (100%) |
| Subject4 | 26/40 (65%) | 25/40 (62.5%) | 39/40 (97.5%) | 34/40 (85%) | 40/40 (100%) | 39/40 (97.5%) |
| Subject5 | 18/40 (45%) | 28/40 (70%) | 35/40 (87.5%) | 31/40 (77.5%) | 39/40 (97.5%) | 40/40 (100%) |
| Subject6 | 25/40 (62.5%) | 27/40 (67.5%) | 33/40 (82.5%) | 35/40 (87.5%) | 40/40 (100%) | 40/40 (100%) |
| Subject7 | 22/40 (55%) | 25/40 (62.5%) | 27/40 (67.5%) | 33/40 (82.5%) | 39/40 (97.5%) | 39/40 (97.5%) |
| Subject8 | 22/40 (55%) | 30/40 (75%) | 37/40 (92.5%) | 33/40 (82.5%) | 39/40 (97.5%) | 40/40 (100%) |
| Subject9 | 26/40 (65%) | 32/40 (80%) | 34/40 (85%) | 39/40 (97.5%) | 40/40 (100%) | 40/40 (100%) |
| Subject10 | 26/40 (65%) | 30/40 (75%) | 35/40 (87.5%) | 35/40 (87.5%) | 40/40 (100%) | 40/40 (100%) |
| **Mean** | **23.4/40 (58.5%)** | **27.4/40 (69.4%)** | **33.8/40 (84.5%)** | **34.5/40 (86.2%)** | **39.7/40 (99.2%)** | **39.7/40 (99.2%)** |

**Table 5. Questionary and answers.**

| Question | S1 | S2 | S3 | S4 | S5 | S6 | S7 | S8 | S9 | S10 | Proposed Paradigm Ratio |
|---|---|---|---|---|---|---|---|---|---|---|---|
| **Q1.** In which paradigm you were more mentally tired? | 2D | 3D | 2D | 2D | 2D | 2D | 2D | 2D | 2D | 2D | 1/10 |
| **Q2.** In which paradigm did you count flashes more easily? | 3D | 3D | 3D | 3D | 3D | 3D | 3D | 3D | 3D | 3D | 10/10 |
| **Q3.** In which paradigm were your eyes more tired? | 2D | 2D | 2D | 2D | 2D | 2D | 2D | 2D | 2D | 2D | 0/10 |
| **Q4.** In which paradigm did you get more distracted? | 2D | 2D | 2D | 2D | 2D | 2D | 2D | 2D | 2D | 2D | 0/10 |
| **Q5.** In which paradigm did you feel more comfortable in general? | 3D | 3D | 3D | 3D | 3D | 3D | 3D | 3D | 3D | 3D | 10/10 |

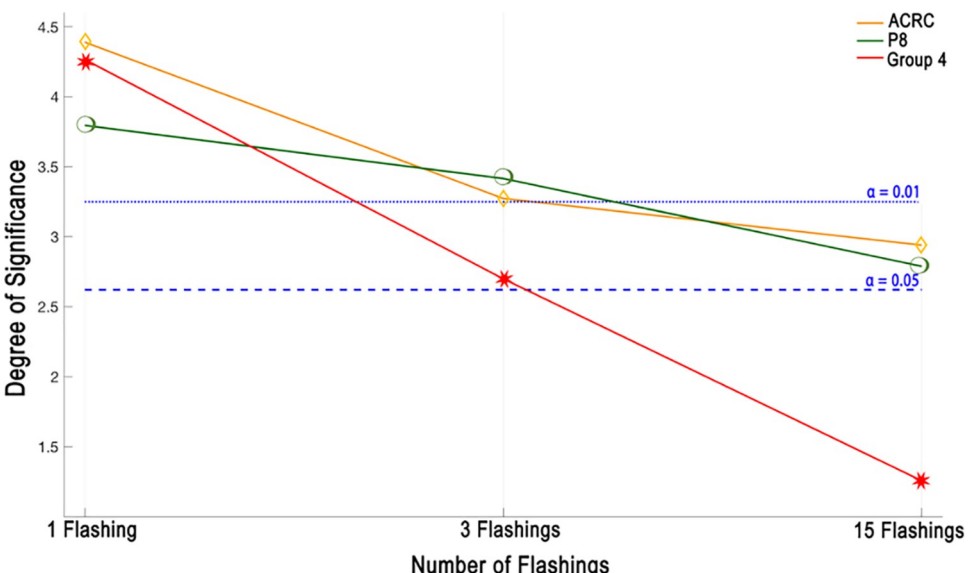

**Fig 15. Degree of significant results in terms of group EEG electrodes according to different number of flashing classification accuracy results.**

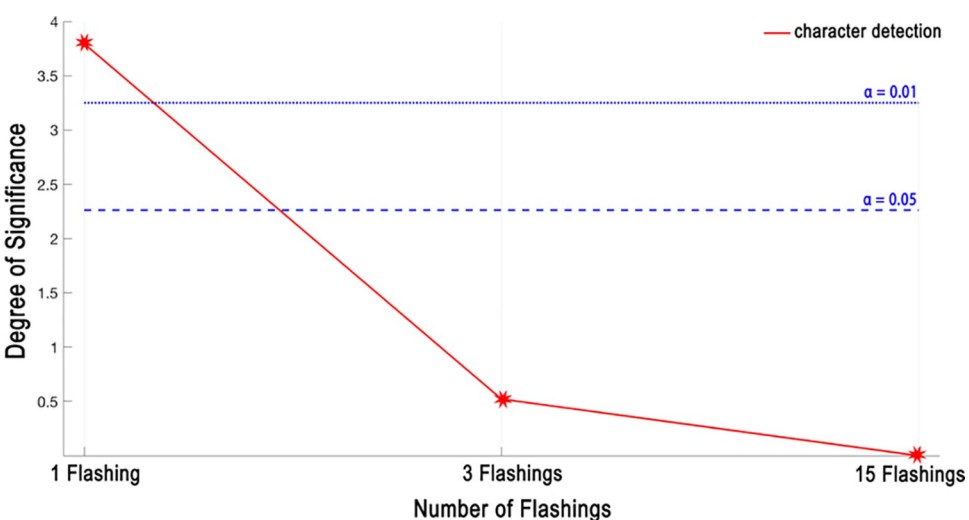

**Fig 16. Degree of significance results in character detection.**

Finally, the subjective test results suggest that majority of the participants prefer the use of the proposed paradigm instead of the classical ones. Considering all these results, the proposed paradigm is a good candidate for next-generation BCI systems that offers simplicity with high accuracy.

## Conclusion

In this study, we proposed a new 3D-C flashing-based P300 speller paradigm, which not only increased the CA rates but also provided a lower user workload. It is shown, in this study, that the proposed speller paradigm is especially useful in BCI systems designed for less EEG electrode usage. It is also observed that the proposed paradigm performs better with fewer flashings compared to the classical 2D-RC paradigm. Especially, the best average CA performance for P8 electrode for 1, 3, and 15 flashings improved to 9.69%, 4.72%, and 1.73%, respectively. Hence, the improvement rate for single flashing, which is important for a BCI system design with low computation time, is considerably high. On the other hand, the best average CA performance for Group 4 electrodes for 1, 3, and 15 flashings improved to 4.36%, 1.01%, and 0.11%, respectively, but they are less compared to single electrode improvement rates. This is because such paradigms reach the maximum accuracy performance (100%) when the number of EEG electrodes or number of flashings is increased. The significance test is also used to evaluate the performance increase, and it is observed that the proposed paradigm improves the performance statistically more significantly with less electrodes and flashings. We believe that the proposed method has a great potential to achieve higher performance for improving the BCI spelling systems. On the other hand, we make the entire dataset publicly available to the community to encourage the reproducibility of this work.

## Supporting information

**S1 File.**
(DOCX)

## Acknowledgments

The authors thank all the participants who took part in the experiment.

## Author Contributions

**Data curation:** Onur Erdem Korkmaz.

**Investigation:** Onur Erdem Korkmaz.

**Methodology:** Onur Erdem Korkmaz, Emin Argun Oral.

**Project administration:** Ibrahim Yucel Ozbek.

**Resources:** Onur Erdem Korkmaz.

**Software:** Onur Erdem Korkmaz.

**Supervision:** Onder Aydemir, Emin Argun Oral, Ibrahim Yucel Ozbek.

**Validation:** Onur Erdem Korkmaz.

**Visualization:** Onur Erdem Korkmaz.

**Writing – original draft:** Onur Erdem Korkmaz.

**Writing – review & editing:** Onder Aydemir, Emin Argun Oral, Ibrahim Yucel Ozbek.

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
