## [Decision Letter · Decision Letter 0]

7 Jan 2022

PONE-D-21-29800An efficient 3D column-only P300 speller paradigm utilizing few numbers of electrodes and flashings for practical BCI implementationPLOS ONE

Dear Dr. korkmaz,

Thank you for submitting your manuscript to PLOS ONE. After careful consideration, we feel that it has merit but does not fully meet PLOS ONE’s publication criteria as it currently stands. Therefore, we invite you to submit a revised version of the manuscript that addresses the points raised during the review process.

ACADEMIC EDITOR: Ensure the quality and legibility of all figures;use the citation whenever required;include the contribution in bullet format==============================

We look forward to receiving your revised manuscript.

Kind regards,

M. Shamim Kaiser, PhD

Academic Editor

PLOS ONE

https://journals.plos.org/plosone/s/file?id=ba62/PLOSOne_formatting_sample_title_authors_affiliations.pdf”

“No”

“NO authors have competing interests”

Reviewers' comments:

Reviewer's Responses to Questions

**Comments to the Author**

1. Is the manuscript technically sound, and do the data support the conclusions?

Reviewer #1: Partly

Reviewer #2: Yes

2. Has the statistical analysis been performed appropriately and rigorously? 

Reviewer #1: Yes

Reviewer #2: Yes

3. Have the authors made all data underlying the findings in their manuscript fully available?

Reviewer #1: No

Reviewer #2: Yes

4. Is the manuscript presented in an intelligible fashion and written in standard English?

Reviewer #1: Yes

Reviewer #2: Yes

5. Review Comments to the Author

Reviewer #1: Please amend the title either on the online submission form or in your manuscript so that they are identical. Please explain the rationale for the development of your work, and clearly indicating which problem with existing approaches you are addressing. Also describe about the datasets used in this work briefly. Please clearly report at the beginning of your methods or results section which the key performance measures were to establish validity and utility of your new system.

Reviewer #2: In Brain Computer Interfaces (BCI), an event of P300 potentials, positive waveforms in electroencephalography (EEG) signals are used often. There are many studies in this field where main focus is to improve performance.

Special thanks to authors for choosing this kind of application oriented field.

However,

1. Figure 1 should be clearer.

2. Data set references are needed.

6. PLOS authors have the option to publish the peer review history of their article (what does this mean?). If published, this will include your full peer review and any attached files.

Reviewer #1: No

Reviewer #2: No

---

## [Author Response · Author response to Decision Letter 0]

15 Jan 2022

AUTHOR RESPONSES TO THE REVIEWERS

Journal: PONE-D-21-29800

Title of the manuscript: An efficient 3D column-only P300 speller paradigm utilizing few numbers of electrodes and flashings for practical BCI implementation

Authors: Onur Erdem Korkmaz, Onder Aydemir, Emin Argun Oral and Ibrahim Yucel Ozbek

We would like to start by thanking the editor(s) and the reviewers for their useful comments and suggestions on how to further improve the quality of my manuscript. I have been able to incorporate changes to reflect all the suggestions provided by the reviewer. Here is a point-by-point response to the reviewer’s comments and concerns.

Reviewer 1:

1. Asked: Please amend the title either on the online submission form or in your manuscript so that they are identical. 

1. Answer: We amended the titles both on the online submission form and in our manuscript.

2. Asked: Please explain the rationale for the development of your work, and clearly indicating which problem with existing approaches you are addressing. Also describe about the datasets used in this work briefly. Please clearly report at the beginning of your methods or results section which the key performance measures were to establish validity and utility of your new system.

2. Answer: We are very grateful to reviewer for his valuable comment, which helped us to improve the quality of our manuscript. Based on his/her suggestion we explain the important points, including the problem which we addressed, dataset and achieved performance, in the subsection of The Proposed Method. The added sentences are also given below.

In this study, we proposed a 3D column (3D-C) intensified based P300 speller system, which provided higher CA rate and lower user workload than the classical 2D-RC P300 speller paradigm. The EEG signals were recorded in both 3D and 2D paradigm procedures from ten healthy participants. In both procedures, 60 characters were presented to each participant during the data collection phase. The proposed 3D-C paradigm was successfully applied to the datasets and we achieved an average CA rate of 99.81% for binary classification (target-nontarget) and 99.2% character detection accuracy on the test data by the traditional two-layer artificial neural networks (ANN) model with a single output neuron.

Please see The Proposed Method section on page 4.

Moreover, we added a brief introduction at the beginning of The Results section as follows:

In this study, we proposed an efficient 3D column-only P300 speller paradigm utilizing few numbers of electrodes and flashings for practical BCI implementation. In this section, we provided the experimental results including P300 detection and the target character recognition performances achieved by the traditional two-layer ANN model with a single output neuron.

Please see the Results section on page 11.

Reviewer 2:

1. Asked: Figure 1 should be clearer.

1. Answer: Based on the reviewer’s suggestion, we improved the readability of Figure 1.

2. Asked: Data set references are needed.

2. Answer: We are very grateful to the reviewer pointing out this issue. The datasets presented in this study were used for the first time in the literature. Therefore, this paper will be cited by the researchers, who will use these datasets. On the other hand, in the section of Introduction, we cited other datasets which were presented by other researchers. 

Comment from the editor:

Asked: The PLOS Data policy requires authors to make all data underlying the findings described in their manuscript fully available without restriction, with rare exception (please refer to the Data Availability Statement in the manuscript PDF file). The data should be provided as part of the manuscript or its supporting information, or deposited to a public repository. For example, in addition to summary statistics, the data points behind means, medians and variance measures should be available. If there are restrictions on publicly sharing data—e.g. participant privacy or use of data from a third party—those must be specified.

Answer: We mentioned the Data Availability in the section of Acknowledgement as follows:

On the other hand, the datasets, which were presented in this study, could be available for the researchers by contacting the corresponding author via e-mail.

Please see the section of Acknowledgement on page 15.

---

## [Decision Letter · Decision Letter 1]

18 Feb 2022

PONE-D-21-29800R1An Efficient 3D Column-Only P300 Speller Paradigm Utilizing Few Numbers of Electrodes and Flashings for Practical BCI ImplementationPLOS ONE

Dear Dr. korkmaz,

Thank you for submitting your manuscript to PLOS ONE. After careful consideration, we feel that it has merit but does not fully meet PLOS ONE’s publication criteria as it currently stands. Therefore, we invite you to submit a revised version of the manuscript that addresses the points raised during the review process.

ACADEMIC EDITOR: Please check the title of the paper in the systemThere are typos in the article, please read the paper very carefully. The font size in the figures must be same. Also mention the impact and reproducibility of this work.Please submit your revised manuscript by Apr 04 2022 11:59PM. If you will need more time than this to complete your revisions, please reply to this message or contact the journal office at plosone@plos.org. Please include the following items when submitting your revised manuscript:A rebuttal letter that responds to each point raised by the academic editor and reviewer(s). You should upload this letter as a separate file labeled 'Response to Reviewers'.A marked-up copy of your manuscript that highlights changes made to the original version. You should upload this as a separate file labeled 'Revised Manuscript with Track Changes'.An unmarked version of your revised paper without tracked changes. You should upload this as a separate file labeled 'Manuscript'.If applicable, we recommend that you deposit your laboratory protocols in protocols.io to enhance the reproducibility of your results. Protocols.io assigns your protocol its own identifier (DOI) so that it can be cited independently in the future. For instructions see: https://journals.plos.org/plosone/s/submission-guidelines#loc-laboratory-protocols. Additionally, PLOS ONE offers an option for publishing peer-reviewed Lab Protocol articles, which describe protocols hosted on protocols.io. Read more information on sharing protocols at https://plos.org/protocols?utm_medium=editorial-email&utm_source=authorletters&utm_campaign=protocols.

We look forward to receiving your revised manuscript.

Kind regards,

M. Shamim Kaiser, PhD

Academic Editor

PLOS ONE

Journal Requirements:

Reviewers' comments:

Reviewer's Responses to Questions

**Comments to the Author**

1. If the authors have adequately addressed your comments raised in a previous round of review and you feel that this manuscript is now acceptable for publication, you may indicate that here to bypass the “Comments to the Author” section, enter your conflict of interest statement in the “Confidential to Editor” section, and submit your "Accept" recommendation.

Reviewer #1: All comments have been addressed

Reviewer #2: All comments have been addressed

2. Is the manuscript technically sound, and do the data support the conclusions?

Reviewer #1: Yes

Reviewer #2: Yes

3. Has the statistical analysis been performed appropriately and rigorously? 

Reviewer #1: Yes

Reviewer #2: Yes

4. Have the authors made all data underlying the findings in their manuscript fully available?

Reviewer #1: Yes

Reviewer #2: Yes

5. Is the manuscript presented in an intelligible fashion and written in standard English?

Reviewer #1: Yes

Reviewer #2: Yes

6. Review Comments to the Author

Reviewer #1: Explaination and overall writing is good. Keep doing this type of work and contribute more in this field.

Reviewer #2: Brain computer interface (BCI) is a methodical system that interfaces the brain to a wide variety of electronic

devices including computer, robotic arm, and mobile phone.

Authors did a good work on this arena.

They proposed a new 3D column flashing based P300 speller paradigm (3D-C). It increases 377 the CA rates. Moreover, it provides lower user workload. Their proposed speller paradigm 378 is specially useful in BCI systems designed for less EEG electrode usage. Furthermore, they showed that their proposed

379 paradigm performs better with less flashings compared to the classical 2D-RC paradigm. Significantly, it is mentioned that the best 380 average CA performance for P8 electrode for 1, 3 and 15 flashings improved to 9.69%, 4.72%, and 1.73%, 381 respectively.

7. PLOS authors have the option to publish the peer review history of their article (what does this mean?). If published, this will include your full peer review and any attached files.

Reviewer #1: No

Reviewer #2: No

---

## [Author Response · Author response to Decision Letter 1]

19 Feb 2022

AUTHOR RESPONSES TO THE ACADEMIC EDITOR

Journal: Plos One

Title of the manuscript: An Efficient 3D Column-Only P300 Speller Paradigm Utilizing Few Numbers of Electrodes and Flashings for Practical BCI Implementation

Manuscript ID: PONE-D-21-29800R1

Authors: Onur Erdem Korkmaz, Onder Aydemir, Emin Argun Oral and Ibrahim Yucel Ozbek

We would like to start by thanking the editor(s) and the reviewers for their useful comments and suggestions on how to further improve the quality of my manuscript. We have been able to incorporate changes to reflect all the suggestions provided by the academic editor. Here is a point-by-point response to the academic editor.

Academic Editor:

1. Asked: Please check the title of the paper in the system.

1. Answer: We checked the title of the paper in the system. We realized that while the first letters of the words were not written in capital letters in the manuscript, but they were given with capital letters in the system. Therefore, we edited the manuscript with capital letters.

2. Asked: There are typos in the article, please read the paper very carefully.

2. Answer: We read the paper very carefully and corrected the typos.

3. Asked: The font size in the figures must be same.

3. Answer: We set the font sizes same in the figures.

4. Asked: Also mention the impact and reproducibility of this work.

4. Answer: We mentioned the impact and reproducibility of this work at the end of the Conclusion section. The added statement is also given below:

We believe that the proposed method has a great potential to achieve higher performance for improving the BCI spelling systems. On the other hand, we make the entire dataset publicly available to the community to encourage the reproducibility of this work.

---

## [Editor Report · Decision Letter 2]

10 Mar 2022

An Efficient 3D Column-Only P300 Speller Paradigm Utilizing Few Numbers of Electrodes and Flashings for Practical BCI Implementation

PONE-D-21-29800R2

Dear Dr. korkmaz,

We’re pleased to inform you that your manuscript has been judged scientifically suitable for publication and will be formally accepted for publication once it meets all outstanding technical requirements.

Kind regards,

M. Shamim Kaiser, PhD

Academic Editor

PLOS ONE

Additional Editor Comments (optional):

Please read the article very carefully to eliminate typos. 
---

## [Editor Report · Acceptance letter]

4 Apr 2022

PONE-D-21-29800R2 

An Efficient 3D Column-Only P300 Speller Paradigm Utilizing Few Numbers of Electrodes and Flashings for Practical BCI Implementation. 

Dear Dr. Korkmaz:

I'm pleased to inform you that your manuscript has been deemed suitable for publication in PLOS ONE. Congratulations! Your manuscript is now with our production department. 

Kind regards, 

on behalf of

Dr. M. Shamim Kaiser 

Academic Editor

PLOS ONE